# Can Alveolar-Arterial Difference and Lung Ultrasound Help the Clinical Decision Making in Patients with COVID-19?

**DOI:** 10.3390/diagnostics11050761

**Published:** 2021-04-23

**Authors:** Gianmarco Secco, Francesco Salinaro, Carlo Bellazzi, Marco La Salvia, Marzia Delorenzo, Caterina Zattera, Bruno Barcella, Flavia Resta, Giulia Vezzoni, Marco Bonzano, Giovanni Cappa, Raffaele Bruno, Ivo Casagranda, Stefano Perlini

**Affiliations:** 1Emergency Medicine Unit and Emergency Medicine Postgraduate Training Program, Department of Internal Medicine, University of Pavia, IRCCS Policlinico San Matteo Foundation, 27100 Pavia, Italy; f.salinaro@smatteo.pv.it (F.S.); carlo.bellazzi01@universitadipavia.it (C.B.); marzia.delorenzo01@universitadipavia.it (M.D.); caterina.zattera01@universitadipavia.it (C.Z.); bruno.barcella01@universitadipavia.it (B.B.); flavia.resta01@universitadipavia.it (F.R.); giuliamaria.vezzoni01@universitadipavia.it (G.V.); marco.bonzano@virgilio.it (M.B.); giovanni.cappa01@universitadipavia.it (G.C.); 2Department of Electrical, Computer and Biomedical Engineering, University of Pavia, 27100 Pavia, Italy; marco.lasalvia01@universitadipavia.it; 3Infectious Disease Unit, Department of Internal Medicine, University of Pavia, IRCCS Policlinico San Matteo Foundation, 27100 Pavia, Italy; raffaele.bruno@unipv.it; 4Academy of Emergency Medicine and Care (AcEMC), 27100 Pavia, Italy; ivo.casagrada@unipv.it

**Keywords:** COVID-19, arterial-alveolar difference, lung ultrasound, P/F, pneumonia, lung injury, emergency department

## Abstract

Background: COVID-19 is an emerging infectious disease, that is heavily challenging health systems worldwide. Admission Arterial Blood Gas (ABG) and Lung Ultrasound (LUS) can be of great help in clinical decision making, especially during the current pandemic and the consequent overcrowding of the Emergency Department (ED). The aim of the study was to demonstrate the capability of alveolar-to-arterial oxygen difference (AaDO_2_) in predicting the need for subsequent oxygen support and survival in patients with COVID-19 infection, especially in the presence of baseline normal PaO_2_/FiO_2_ ratio (P/F) values. Methods: A cohort of 223 swab-confirmed COVID-19 patients underwent clinical evaluation, blood tests, ABG and LUS in the ED. LUS score was derived from 12 ultrasound lung windows. AaDO_2_ was derived as AaDO_2_ = ((FiO_2_) (Atmospheric pressure − H_2_O pressure) − (PaCO_2_/R)) − PaO_2_. Endpoints were subsequent oxygen support need and survival. Results: A close relationship between AaDO_2_ and P/F and between AaDO_2_ and LUS score was observed (R^2^ = 0.88 and R^2^ = 0.67, respectively; *p* < 0.001 for both). In the subgroup of patients with P/F between 300 and 400, 94.7% (*n* = 107) had high AaDO_2_ values, and 51.4% (*n* = 55) received oxygen support, with 2 ICU admissions and 10 deaths. According to ROC analysis, AaDO_2_ > 39.4 had 83.6% sensitivity and 90.5% specificity (AUC 0.936; *p* < 0.001) in predicting subsequent oxygen support, whereas a LUS score > 6 showed 89.7% sensitivity and 75.0% specificity (AUC 0.896; *p* < 0.001). Kaplan–Meier curves showed different mortality in the AaDO_2_ subgroups (*p* = 0.0025). Conclusions: LUS and AaDO_2_ are easy and effective tools, which allow bedside risk stratification in patients with COVID-19, especially when P/F values, signs, and symptoms are not indicative of severe lung dysfunction.

## 1. Introduction

Coronavirus disease 2019 (COVID-19) caused by severe acute respiratory syndrome coronavirus-2 (SARS-CoV-2) has posed an unprecedented challenge to global health systems. The World Health Organization (WHO) has declared coronavirus disease 2019 (COVID-19) a public health emergency of international concern, with numbers and geography of a real pandemic [1]. COVID-19 infection can be associated with radiological diagnosis of interstitial pneumonia and alteration in gas exchange. Patients with severe infection frequently present arterial hypoxemia and progress to acute respiratory distress syndrome (ARDS) requiring intensive care unit (ICU) admission and mechanical ventilation [2] (approximately 5–10% of cases). In respiratory diseases, a key role is played by data provided by arterial blood gas (ABG) and lung ultrasound (LUS) [3]. They orient the early diagnosis and severity stratification of the disease, allowing provision of early and adequate therapy [4]. Two commonly used indices to evaluate the pathogenic mechanism of respiratory failure and its severity are the PaO_2_ / FiO_2_ ratio (P/F) and the alveolar-to-arterial oxygen difference (AaDO_2_) [5]. While P/F can be used in the clinical practice as a simple measure of lung dysfunction in critically ill patients to predict disease outcome, as highlighted by the Berlin criteria in ARDS patients [6], an elevated AaDO_2_ accompanied by hypoxemia indicates ventilation/perfusion mismatch or intra-pulmonary shunting [7]. COVID-19 pneumonia is associated with increased shunt and/or altered oxygen alveolar–arteriolar barrier diffusion. This might be associated with increased AaDO_2_ and decreased P/F values [7]. The primary aim of the present study is to demonstrate the capability of baseline AaDO_2_ in predicting both the need for oxygen support and survival in patients with COVID-19 infection, as obtained at the time of Emergency Department (ED) admission. Given the recognized role of LUS in this setting [4,8,9], a secondary aim was to evaluate the correlation between AaDO_2_ and LUS results, especially in patients with normal P/F values, since these are these patients who are at higher risk of being underestimated and undertriaged, who might subsequently undergo rapid worsening due to a relatively unexpected clinical evolution. Indeed, simple prognostic indexes are needed to better orient clinical decision-making and safe discharge policy, especially in an overcrowded ED because of the pandemic.

## 2. Materials and Methods

The study enrolled consecutive patients with swab-confirmed COVID-19, from March 2nd to April 22th, 2020. A positive result of high throughput sequencing or real-time reverse-transcriptase–polymerase-chain-reaction (RT-PCR) assay of nasal and pharyngeal swab was the fundamental requirement to be included in the final analysis. After having obtained written informed consent, all patients underwent lung ultrasound, associated with a pre-specified “suspected COVID-19” laboratory test profile, including complete blood count, assessment of renal and liver function, Troponin I, serum electrolytes, C-reactive protein, lactate dehydrogenase, and creatinine kinase. Upon ED admission, vital signs, presentation symptoms, and ABG samples were also collected. AaDO_2_ was calculated relying on the following mathematical formula [5]:AaDO_2_ = ((FiO2) (Atmospheric pressure − H_2_O pressure) − (PaCO_2_/R)) − PaO_2_.We considered standard values for all patients:Atmospheric pressure = 760 mmHgH_2_O pressure = 47 mmHgRespiratory quotient (R) = 0.8Normal values of AaDO_2_ were considered, according to the following formula:Normal AaDO_2_ = 2.5 + 0.21 × age in years [10]

ABG samples were analyzed on Radiometer ABL 825 (Radiometer Medical ApS, Åkandevej 21, DK-2700, Brønshøj, Denmark). Per protocol, while waiting for the swab results, all patients underwent bedside LUS evaluation with Aloka Arietta V70 (Hitachi Medical Systems S.p.A., via Lomellina 27a, I-20090 Buccinasco, Italy, equipped with a convex 5 MHz probe) [11]. The thorax was studied with the patient in the supine or semi-supine position, depending on the level of cooperation. According to guidelines in the emergency setting, LUS examination was conducted by trained ED physicians (experienced sonographers according to the American College of Emergency Physicians ultrasonographic guidelines; more than 10 ultrasound exams performed per week, 5 years of experience in performing and interpreting POCUS) [12] using 12 windows (2 anterior, 2 lateral, and 2 posterior zones per hemithorax) [13,14].

Videoclips were recorded, ensuring analysis throughout the respiratory cycle, to allow subsequent off-line re-evaluation. In each region, a quantitative LUS score was attributed by an external reader, who was blinded to the clinical presentation, as follows: score 0: normal lung aeration (A lines or less than 2 small vertical artifacts); score 1: mild loss of aeration (presence of vertical artifacts or lung consolidation in less than 50% of the pleural line); score 2: severe loss of aeration (“white lung” or coalescent B vertical artifacts or presence of vertical artifacts/lung subpleural consolidation in more than 50% of the pleural line); score 3: complete loss of aeration (predominant tissue-like pattern) [15,16]. Global LUS score was computed as the sum of each regional scores. A prevalent LUS pattern was assigned depending on the presence of only interstitial syndrome (“Interstitial Pattern”), or evidence of subpleural consolidations in at least 2 lung fields (“Consolidation Pattern”), in which the presence of vertical artifacts also coexisted. The absence of lung injury was defined as a LUS score = 0. We considered “Critical Patients” the subjects requiring CPAP/NIV or orotracheal intubation (IOT) and/or admission in Intensive Care Unit (ICU). The relationship between LUS score and ABG respiratory parameters was evaluated in the whole group as well as in the group of patients with P/F 300–400. Statistical analysis relied on MEDCALC 19.2.3 (Ostend, Belgium). Continuous variables were expressed as median values, while categorical variables were expressed as percentages. A *p* < 0.05 value was considered statistically significant. Scatter diagrams, ANOVA, regressions, Kaplan–Meier curves and Receptor Operating Characteristic (ROC) curves, and χ2 analyses were used, as appropriate. No imputation was made for missing data. Because the cohort of patients in our study was not derived from random selection, all statistics are deemed to be descriptive only. Prognosis was censored at 30-days through medical records for hospitalized patients and through phone calls for discharged subjects.

## 3. Results

Out of 820 patients admitted in ED during the observation period, 530 had a SARS-CoV2 positive nasopharyngeal swab. Among them, 223 presented a complete LUS examination and an ABG on room air. Table 1 summarizes the baseline characteristics of complete cases and the different study groups. The median age of patients was 61 years (range 22–90 years) and 61.9% were male; the most frequent presentation symptom was fever (89.7%), followed by cough (48%), and dyspnoea (46.2%). The remaining presenting symptoms were asthenia (13.5%), diarrhea (11.7%), chest pain (9.4%), and confusion (2.3%). A total of 136 (61%) patients had at least one comorbidity (Table 2), and 10.3% of them had 3 or more diseases, hypertension being the most commonly observed (45%), followed by diabetes (14.4%), coronary artery disease (12.6%), asthma (6.3%), chronic kidney disease (4.5%), active cancer (4.1%), and neurological disease (3.6%). Out of 223, 17 patients (7.6%) were admitted to intensive care unit (ICU), 101 (45.3%) in a general ward, while 102 subjects (45.7%) were discharged at home and 3 (1.3%) died in ED. The 23.3% of 223 patients received higher intensity care with CPAP or IOT. Median LUS score was 9 and only 36 patients (16.1%) did not have lung involvement, whereas 100 (44.8%) presented only vertical artifacts and 87 (39.1%) presented both vertical artifacts and consolidations. As to the arterial blood gas results, median pH was 7.45 (7.32–7.60), pCO_2_ 33.5 mmHg (18.6–52.0), pO_2_ 70 mmHg (31–123), P/F 333 (148–586), AaDO_2_ 38.6 mmHg (0.5 to 81). A reduction of P/F and pO_2_ values was related to increasing of severity of the clinical picture. Conversely, AaDO_2_ increased with worsening clinical presentation, median values being 34 and 55 in non-critically ill and critically ill patients, respectively (*p* < 0.001). Figure 1 shows the relationship between AaDO_2_ and P/F and between AaDO_2_ and LUS score (R^2^ = 0.88, *p* < 0.001 and R^2^ = 0.67, *p* < 0.001). Stratifying the patient cohort according to P/F values, as shown in Figure 2, AaDO_2_ increased with decreasing P/F (*p* < 0.001). Analyzing the subgroup of patients with P/F between 300 and 400 (*n* = 113), the median age was 61.5 years (28–90), with an upper calculated reference limit value [10] of AaDO_2_ equal to 21.4 mmHg. Figure 3 shows that within this subgroup 107/113 subjects had a value of AaDO_2_ above the upper calculated reference limit, whereas only the remaining 6 patients were under the upper calculated reference limit [10]. Out of these 107 patients with values above the upper calculated reference limit [10], the 55 who subsequently needed oxygen therapy had higher AaDO_2_ (41.9 ± 6.4 vs 32.9 ± 6.8; *p* < 0.001). Out of these 55, 2 were admitted to the ICU and 10 died (median age of survivors being 60 ± 14 vs 71 ± 16 years; *p* = 0.022); 9 patients died in General Ward (GW) and 1 died in ICU. In contrast, only 1 out of 6 patients under the upper calculated reference limit needed oxygen support. In the subgroup with P/F between 300 and 400, patients who subsequently needed oxygen support had higher LUS score (10 ± 3.8 vs 6 ± 3.9; *p* < 0.001). When comparing the AaDO_2_ values with the ultrasound pattern, patients with a defined “Consolidation Pattern” had higher AaDO_2_ values when compared with patients with either an “Interstitial Pattern” or the absence of pulmonary involvement (AaDO_2_ value: 45.3 ± 14 vs 39.2 ± 14 vs 15.2 ± 11, respectively; *p* < 0.001). It is interesting to note that, among the 102 patients discharged at home, only 10 returned to the ED in the following 30 days for problems related to COVID-19 diagnosis. In particular, during the first presentation in ED, 3 of them had a LUS score > 6, and 8 had an increased AaDO_2_. Notably, none of them had a P / F value < 330. According to ROC curve analysis on the whole cohort (Figure 4), AaDO_2_ > 39.4 had 83.6% sensitivity and 90.5% specificity, with 90.7% positive predictive value (PPV) and 83.5% negative predictive value (NPV) in predicting the need for high flow of oxygen, whereas AaDO_2_ > 57.2 had 46.9% sensitivity and 90.7% specificity in predicting death at 30 days (AUC = 0.936 and AUC = 0.744, *p* < 0.0001). Similar results were obtained on the subgroup of patients with P/F 300–400; AaDO_2_ > 36.4 had 78.6% sensitivity and 75.4% specificity in predicting the need for high flow of oxygen (AUC = 0.831, *p* < 0.001). The subsequent need for oxygen support was also predicted by LUS score > 6 with 89.7% sensitivity and 75% specificity (AUC 0.896; *p* < 0.001), with 80% positive predictive value (PPV) and 86.7% negative predictive value (NPV). Survival was also predicted by AaDO_2_, as shown by Kaplan–Meier analysis (Figure 5).

## 4. Discussion

Based on an observational cohort of 223 consecutive COVID-19 patients, evaluated at San Matteo University Hospital in Pavia (Italy), the present study shows as its main result that AaDO_2_ can be a useful parameter to stratify the evolutionary risk of patients with COVID-19 [17]. To the best of our knowledge, this is the first paper that evaluates the potential role of AaDO_2_, as derived by admission ABG, for a better characterization of COVID-19 patients. ABG is easily available in the emergency setting, immediately giving crucial information about pulmonary involvement and respiratory function. Although P/F ratio has gained a larger popularity [7] as a simple measure of pulmonary dysfunction in critically ill patients, AaDO_2_ enables more a precise evaluation of the pathophysiological basis of hypoxemia. In particular, a high AaDO_2_ value associated with normal or low pCO_2_ means either ventilation/perfusion mismatch or intrapulmonary shunting [18]. The strong correlation between AaDO_2_, P/F, and LUS score that was observed in the present study demonstrates that alveolar-to-arterial oxygen difference can be a reliable measure of pulmonary dysfunction. Moreover, the combination of an imaging finding such as the LUS score, which is able to provide a quantitative/qualitative estimate of lung involvement, with a respiratory parameter such as AaDO_2_, allows a better understanding of the underlying mechanism. Patients with a “Consolidative Pattern”, i.e., with lower lung aeration, had higher AaDO_2_ values when compared with patients with “Interstitial Syndrome Pattern” or the absence of lung involvement as inferred from lung ultrasound. This may represent a consequence of greater intrapulmonary shunting or ventilation/perfusion mismatch. The role of AaDO_2_ in stratifying the risk in lung diseases has been reported in patients hospitalized with community-acquired pneumonia [19,20]. The present study evaluated the role of the alveolar-to-arterial oxygen difference particularly in COVID-19 patients with P/F between 300 and 400, that according to literature represents a range of values without significant acute lung injury [21]. In this subgroup of patients it has to be noted that despite normal P/F values, AaDO_2_ was increased. Moreover, more than half of these patients did subsequently require oxygen therapy support. Interestingly, patients who subsequently needed oxygen support had a more severe extent of lung involvement, as assessed by LUS, than those who did not. Recently Tobin and coworkers [22] highlighted that patients with COVID-19 pneumonia often do not report dyspnoea, despite extreme hypoxemic values. They defined this clinical presentation as “silent hypoxemia” or “happy hypoxia”, with physical signs that may either overestimate or underestimate patient discomfort [23]. In patients who present with few signs and symptoms, a chest X-ray that is not indicative of significant lung involvement, [24] and P/F still within normal limits, it is of utmost importance to obtain elements that can reliably predict the risk of subsequent clinical worsening. Otherwise, especially in an overcrowded ED, these subjects could be unwisely (and unsafely) discharged. In this subset of patients, LUS and AaDO_2_ can predict the subsequent need for oxygen therapy and can help detection of early lung involvement. It is important to note that, in our series of 107 patients presenting with AaDO_2_ above the upper calculated reference limit value and P/F > 300, more than half (*n* = 55) subsequently needed oxygen support, with 2 ICU admissions and 10 intrahospital deaths. According to ROC curve analysis, AaDO_2_ had high sensitivity and specificity in predicting both oxygen need and 30-day mortality. Moreover, Kaplan–Meier curves show that patients with higher AaDO_2_ values had a lower probability of survival. Again, it is important to point out that these data are derived from patients with P/F values > 300, namely indicating subjects without evident acute lung injury. It is worth underscoring that at variance with the easy-to-calculate P/F ratio [7], AaDO_2_ values does take into account the underlying physiopathological aspects, such as the changes in alveolar–arterial exchange that occur with age [10,25]. Of course, the first ED evaluation intercepts patients at different stages of the disease. For this reason, a more accurate definition of functional characteristics (ABG) and imaging (LUS) of COVID-19 patients is always desirable. The predictive role of AaDO_2_ represents a very powerful tool helping a closer follow-up of subjects at higher risk of the subsequent need for oxygen support despite a milder clinical presentation upon ED admission. A further observation is related to age, since higher mortality was observed in older patients, despite an initially normal P/F ratio. LUS and AaDO_2_ proved to be important predictors of oxygen therapy need and clinical deterioration in a group of patients with normal P/F values and a relatively mild clinical presentation who have a higher probability of being discharged without receiving proper attention, especially in the setting of pandemic-related ED overcrowding. This potential risk can be prevented by combined LUS and AaDO_2_ evaluation, which can be quickly and easily performed at the bedside by the ED team. From an ED doctor’s standpoint, the early recognition of worsening risk is as difficult as important, in order to safely discharge and adequately allocate patients and resources, especially during a pandemic time.

### Study Limitations

Some limitations of this study should be acknowledged. The retrospective single-center design leads to missing data and unavoidable biases in identifying and recruiting participants. The data were obtained in times of health emergency situation, and the sample size was relatively small. Despite these limitations, the study reflects the ‘real life’ clinical situation in the ED during a pandemic outbreak. Despite the encouraging results, further validation is warranted in future multi-center large prospective studies to consolidate the use of LUS and AaDO_2_ evaluation.

## 5. Conclusion

In the interest of guiding clinical decision-making in the setting of an overcrowded ED because of the challenging pandemic of COVID-19, LUS and AaDO_2_ can be easy and effective tools to predict a clinical worsening, especially in the subgroup of patients without a clearcut respiratory failure (P/F > 300). Their routine integration into clinical evaluation of COVID-19 patients is strongly suggested.

## Figures and Tables

**Figure 1 diagnostics-11-00761-f001:**
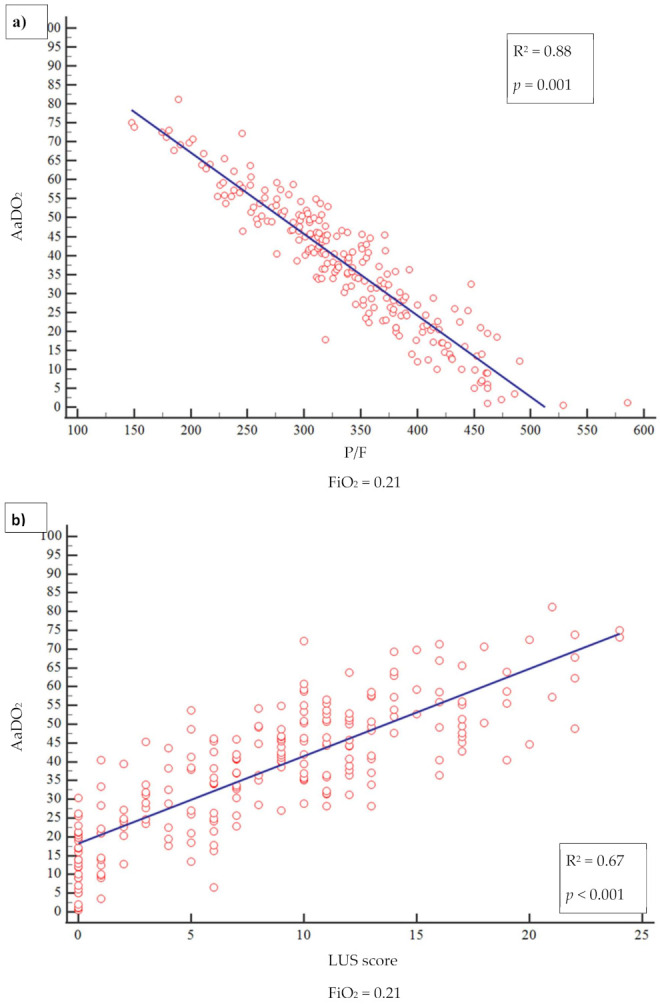
(**a**) relationship between AaDO2 and P/F and (**b**) between AaDO2 and LUS.

**Figure 2 diagnostics-11-00761-f002:**
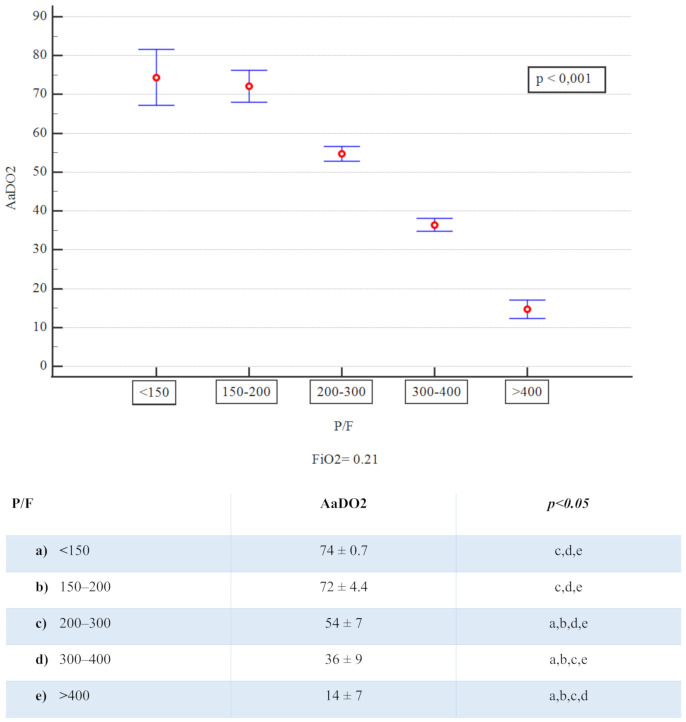
AaDO2 mean values and P/F groups.

**Figure 3 diagnostics-11-00761-f003:**
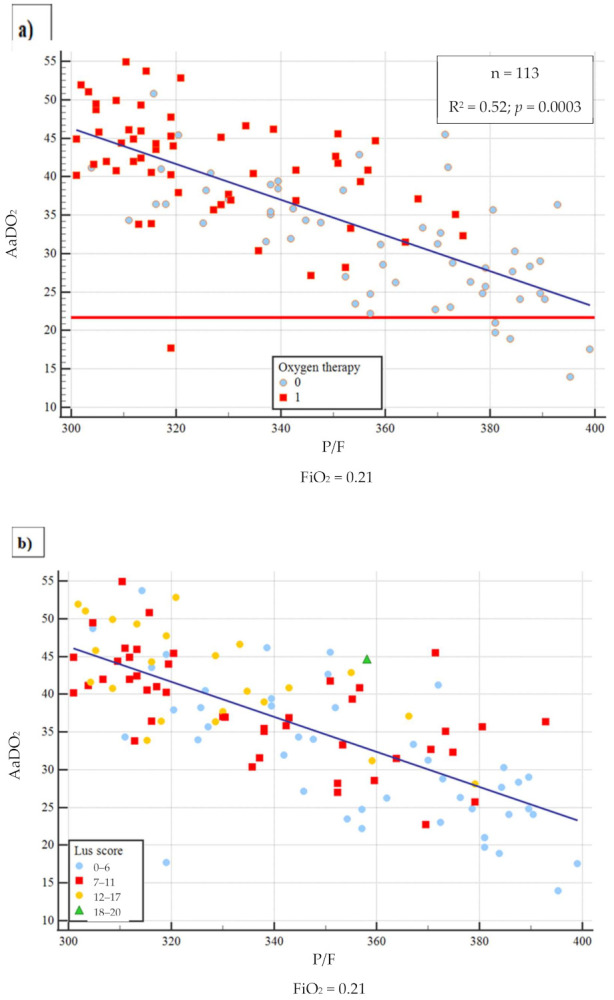
Distribution of AaDO2 values in the P/F 300–400 group. (**a**) Oxygen therapy (**b**) LUS score distribution: blue (0–6), red (7–11), orange (12–17) and green (18–20).

**Figure 4 diagnostics-11-00761-f004:**
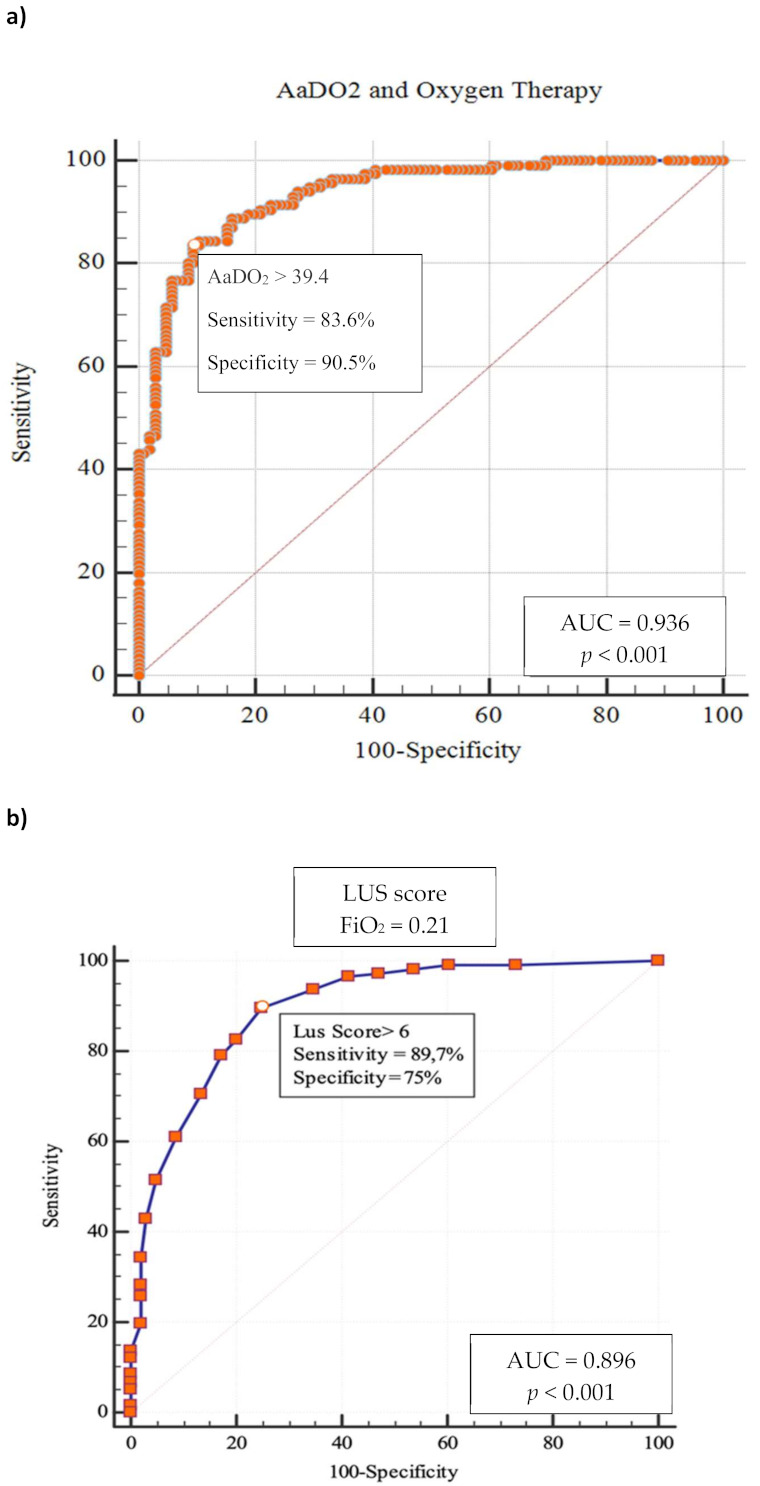
ROC curves in whole cohort. (**a**) AaDO_2_ and Oxygen Therapy. (**b**) LUS score and Oxygen Therapy.

**Figure 5 diagnostics-11-00761-f005:**
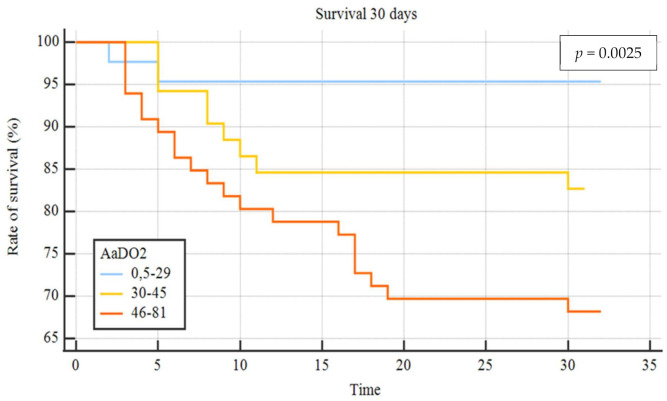
AaDO_2_ Kaplan–Meier curves in whole cohort. AaDO_2_ groups of values: blue line (0.5–29), yellow line (30–45) and orange line (46–81).

**Table 1 diagnostics-11-00761-t001:** Patients baseline characteristics.

	Overall (*n* = 223)	Not Critical Patients (*n* = 171)	Critical Patients (*n* = 52)	*p* Value
Age (years)	61 (22–90)	58 (22–90)	69.5 (42–89)	*p* < 0.001
SEX (male %)	61.9%	57.3%	76.9%	*p* = 0.01
BMI (kg/m^2^)	26.2 (18.6–45.7)	26.2 (18.7–45.7)	26.2 (22.9–40.8)	n.s
Arterial Systolic Pressure (mmHg)	130 (80–190)	130 (90–190)	134 (80–174)	n.s
Arterial Diastolic Pressure (mmHg)	80 (50–117)	80 (50–117)	80 (50–110)	n.s
Heart Rate (bpm)	88 (40–135)	86 (40–130)	91 (60–135)	n.s
Respiratory Rate (/min)	20 (10–44)	18.5 (10–44)	22 (10–40)	*p* = 0.016
CRP (mg/dL)	5.4 (0.01–41.9)	3.36 (0.01–29.3)	14.2 (0.84–41.9)	*p* < 0.001
Hb (g/dL)	13.9 (8.4–23.5)	13.8 (10–23.5)	13.9 (8.4–17.2)	n.s
Lymphocytes (×10^3^/uL)	0.9 (0.1–3.9)	1 (0.1–3.9)	0.8 (0.2–1.9)	*p* = 0.001
LDH (mU/mL)	297 (122–2578)	282 (122–852)	408 (223–2578)	*p* < 0.001
TnI (ng/mL)	7 (2.5–885)	5 (2.5–885)	14.5 (2.5–218)	n.s
CPK (mU/mL)	117 (22–46737)	97 (22–2130)	153 (24–46737)	n.s
Creatinin (mg/dL)	0.85 (0.37–4.4)	0.82 (0.37–3.4)	1.04 (0.56–4.4)	*p* < 0.001
PaO_2_/FiO_2_	333 (148–586)	352 (191–586)	257 (148–375)	*p* < 0.001
AaDO_2_	38.6 (0.5–81)	34 (0.5–69)	55 (18–81)	*p* < 0.001
PaCO_2_ (mmHg)	33.5 (18.6–52)	34 (19–43)	31 (21–52)	*p* = 0.003
PaO_2_ (mmHg)	70 (31–123)	74 (40–123)	54 (31–79)	*p* < 0.001
LUS Score	9 (0–24)	6 (0–19)	13.5 (4–24)	*p* < 0.001

**Table 2 diagnostics-11-00761-t002:** Patients comorbidities.

Comorbidity Overall (*n* = 223)	Non Critical Patients (*n* = 171)	Critical Patients (*n* = 52)
Hypertension (45%)	40.9%	57.7%
Diabetes (14.4%)	9.9%	28.8%
CAD (12.6%)	8.2%	26.9%
Asthma (6.3%)	7.6%	1.9%
CKD (4.5%)	4.1%	5.8%
Active Cancer (4.1%)	35.3%	5.8%
Neurological Disease (3.6%)	1.7%	9.6%

CAD: coronary artery disease; CKD: chronic kidney disease.

## Data Availability

Data are available upon reasonable request.

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
