# Peer review of "Can Alveolar-Arterial Difference and Lung Ultrasound Help the Clinical Decision Making in Patients with COVID-19?"

_diagnostics, 2021, doi:10.3390/diagnostics11050761_

Round 1
Reviewer 1 Report
Dear ladies and gentlemen,
thank you very much for the opportunity to review your paper!
Overall I appreciate your valuable work for the assessment of the severity of COVID-19 patients in the Emergency Department.
I have only few remarks which you might want to take into consideration for a revised version:
line 105: And / or admission in ICU – I think it would be fair enough to mention that admission to the intensive care is generally not a standardized procedure. Therefore it can be possible that one physician might have a lower threshold to admit to ICU than another physician unless in the study center there is a standardized protocol available for admission to the ICU.
line 251: expecially -> especially
General comments:
- In crowded Emergency Departments it can be difficult to find time to calculate AaDO2. Therefore it would be advisable to have an easy-to-use approach such as an app for the calculation. The most adequate procedure would be an automated calculation with indication to the treating physician if a pathological result is present. Maybe you want to discuss this for usage in the future.
- Please consider to mention one or both of the following paper since both of them are mentioning the AaDO2 in patients suffering from COVID-19:
- Carlino MV, Valenti N, Cesaro F, Costanzo A, Cristiano G, Guarino M, Sforza A. Predictors of Intensive Care Unit admission in patients with coronavirus disease 2019 (COVID-19). Monaldi Arch Chest Dis. 2020 Jul 15;90(3). doi: 10.4081/monaldi.2020.1410. PMID: 32672430.
- Mohr A, Dannerbeck L, Lange TJ, Pfeifer M, Blaas S, Salzberger B, Hitzenbichler F, Koch M. Cardiopulmonary exercise pattern in patients with persistent dyspnoea after recovery from COVID-19. Multidiscip Respir Med. 2021 Jan 25;16(1):732. doi: 10.4081/mrm.2021.732. PMID: 33623700; PMCID: PMC7893311.
Kind regards,
Author Response
Re: manuscript DIAGNOSTICS-1166543 R1
“CAN ALVEOLAR ARTERIAL DIFFERENCE AND LUNG ULTRASOUND HELP THE CLINICAL DECISION MAKING IN PATIENTS WITH COVID-19?”
Prof. Dr. Andreas Kjaer Pavia, April 8th 2021
Editor-in-Chief
Diagnostics
Dear Prof. Dr. Andreas Kjaer,
enclosed please find the revised version of the manuscript entitled “Can Alveolar Arterial difference and Lung Ultrasound help the clinical decision making in patients with Covid-19?”, together with the point-to-point response to the Reviewers.
We are grateful for the opportunity they gave us to improve the manuscript and to better finalize its take-home message.
The manuscript has not been published, nor is it under consideration elsewhere. None of the paper’s contents have been previously published. None of the figures was ever published elsewhere. No financial or other relationships of any of the Authors might lead to a conflict of interest. The manuscript has been read and approved by all authors, who all have contributed significantly to the submitted work.
We hope you will find the revised manuscript now acceptable for publication in Diagnostics.
With kindest regards,
Yours sincerely,
Stefano Perlini, MD, PhD, FESC
Director, Emergency Medicine Postgraduate Training Program
Internal Medicine, Vascular and Metabolic Disease Unit, Department of Internal Medicine
IRCCS Policlinico San Matteo Foundation, University of Pavia
P.le Golgi, 19 27100 Pavia - Italy
Tel: +39-0382-502568 Fax: +39-0382-502441
email: stefano.perlini@unipv.it
RESPONSE TO THE REVIEWERS
Reviewer #1
Overall I appreciate your valuable work for the assessment of the severity of COVID-19 patients in the Emergency Department. I have only few remarks which you might want to take into consideration for a revised version:
Line 105: And / or admission in ICU – I think it would be fair enough to mention that admission to the intensive care is generally not a standardized procedure. Therefore, it can be possible that one physician might have a lower threshold to admit to ICU than another physician unless in the study center there is a standardized protocol available for admission to the ICU
We gratefully thank the Reviewer for her/his positive and constructive comments, and for finding the manuscript valuable and interesting for the clinician. According to her/his suggestions, we revised the language in order to ameliorate the readability of the paper. We completely agree that admission of a patient to the Intensive Care Unit is generally a non-standardized procedure, although some criteria have to be chosen for the best use of hospital resources. During this challenging pandemic, because of a steadily overcrowded Emergency Department, the admission criterion to ICU was almost exclusively oro-tracheal intubation, whereas patients undergoing noninvasive ventilation were preferentially directed to the Medical Wards (i.e. the Internal Medicine, Respiratory Disease or Infective Disease Units). We clarified this point in the revised manuscript and we thank the Reviewer for having raised this point.
Line 251: expecially -> especially
The typing error was amended, as suggested.
In crowded Emergency Departments it can be difficult to find time to calculate AaDO2. Therefore, it would be advisable to have an easy-to-use approach such as an app for the calculation. The most adequate procedure would be an automated calculation with indication to the treating physician if a pathological result is present. Maybe you want to discuss this for usage in the future.
Since our arterial blood gas analysis machines (Radiometer ABL 825) directly provide AaDO2 values, we did not mention this point as an issue. As correctly pointed out by the Reviewer, some Medical Apps (that can be easily downloaded to any currently used smartphone or a tablet) allow AaDO2 calculation in less than 1 minute. Moreover, age-adjustment, as reported in the text, remains easy-to-use during the data tabulation. We added this suggestion in the revised manuscript and we consider the development of an “alert” system for the attending physician.
Please consider to mention one or both of the following paper since both of them are mentioning the AaDO2 in patients suffering from COVID-19:
- Carlino MV, Valenti N, Cesaro F, Costanzo A, Cristiano G, Guarino M, Sforza A. Predictors of Intensive Care Unit admission in patients with coronavirus disease 2019 (COVID-19). Monaldi Arch Chest Dis. 2020 Jul 15;90(3). doi: 10.4081/monaldi.2020.1410. PMID: 32672430.
- Mohr A, Dannerbeck L, Lange TJ, Pfeifer M, Blaas S, Salzberger B, Hitzenbichler F, Koch M. Cardiopulmonary exercise pattern in patients with persistent dyspnoea after recovery from COVID-19. Multidiscip Respir Med. 2021 Jan 25;16(1):732. doi: 10.4081/mrm.2021.732. PMID: 33623700; PMCID: PMC7893311.
We thank the Reviewer for the suggestion. This add was particularly interesting and useful, since we agree that it is helpful to strengthen our discussion.
We thank the Reviewer for her/his positive and constructive comments, that helped us in revising the manuscript and in improving the final take-home message.
Reviewer 2 Report
Dear authors
Congratulations on your interesting job. Its limitation is certainly the small size of the studied group. However, your calculations are very interesting, the statistical analysis carried out is precise, and the results are interesting. I believe that any study that proves significant correlations between LUS scores and other measurable parameters is valuable. In this context, the relationship you are assessing between the LUS score and AaDO2 is indeed very valuable.
This result alone is worth publishing. Survival analysis is also very interesting - although due to the very small study population, given the overall prevalence of COVID-19, it is difficult to extrapolate this data, despite the temptation. I believe that "restrictions" should be distinguished in the form of a separate paragraph. You have mentioned them, but they are blurred in the discussion.
Table 1 also requires careful checking and correction. Understands that you are giving mean values ? (or medians?) And ranges (or min-max values?). Please also check the data - for example BMI values - mean (?) for the whole group 22.4, and for both subgroups 26.2? No significant differences? Further HR values errors in the values in parentheses: mean 88 (30-40) ??? Please carefully check the values given in the tables and describe the table in detail (which values are average, which medians, or what features of these values are given in parentheses).
I also believe that presenting the general characteristics of the studied population in the context of comorbidities in the form of a table would be clearer.
Finally - despite the valuable results, I have doubts about the final conclusion of the researchers. Do you suggest that every ED patient should have both LUS and AoDO2 assessment? Is the mere assessment of blood gas parameters not enough to qualify a patient as potentially in need of high-flow oxygen therapy? I fully agree with the importance of each of these studies individually - both AaDO2 and LUS - in the initial assessment of a patient with COVID-19 (and more). However, in the context of the chaos in ED highlighted in the discussion, why waste time on both of these studies, since each of them allows patients with silent hypoxemia to "visualize" the true extent of damage to the lungs?
Best regards
Round 2
Reviewer 2 Report
Dear authors
Thank you for considering my comments. Thank you also for the interesting confrontation of points of view in the context of my last comment. I also see that part of the truth is on your side. Perhaps our disagreement on this matter is due to the ED structure we have had the opportunity to work on. However, this does not change the basic point - your work is very interesting, thank you for the opportunity to review it and congratulations once again.
Best Regards
MW